# Cucumber Strigolactone Receptor CsDAD2 and GA₃ Interact to Regulate Shoot Branching in *Arabidopsis thaliana* L.

Yaoliang Cao [1,†], Yanlong Dong [1,2,†], Runming Zhang [1], Qian Li [1], Ruonan Peng [1], Chao Chen [1], Mengdi Lu [1] and Xiaoxia Jin [1,*]

1    "Plant Biology" Key Laboratories of Universities in Heilongjiang Province, College of Life Science and Technology, Harbin Normal University, Harbin 150025, China
2    Horticulture Branch, Heilongjiang Academy of Agricultural Sciences, Harbin 150069, China
\*    Correspondence: xiaoxia6195@126.com
†    These authors contributed equally to this work.

**Abstract:** Previous studies identified that strigolactones (SLs) and gibberellins (GAs) interacted when controlling branching in plant shoots, but the underlying mechanism remains unknown. qRT-PCR analysis suggested that the SL receptor gene *CsDAD2* was significantly upregulated in the leaves, stems, and nodes of cucumber after treatment with 50 mg/L of GA₃. Furthermore, the *CsDAD2* gene was cloned and introduced into wild-type *Arabidopsis* plants via Agrobacterium-mediated transformation. For the CsDAD2-OE lines, the endogenous content of GA₃ was subsequently higher at the seedling stage, with the number of primary cauline branches also significantly increased at the maturity stage compared with WT. Additionally, GA-related genes were up-regulated in the first inter-nodes and the third nodes of the *CsDAD2-OE* lines, thus indicating that GA was metabolically active in these tissues. The expression of the branch inhibitor gene *AtBRC1* decreased at the seedling stage as well as at the maturity stage of the *CsDAD2-OE* lines. These findings suggest that *CsDAD2* might have important functions in the interactions between GAs and SLs as it can promote the accumulation of GAs in plant nodes and suppress the expression of *BRC1*, hence increasing primary cauline branching.

**Keywords:** cucumber; shoot branching; gibberellins; strigolactones; *CsDAD2*

## 1. Introduction

Plant architecture is a crucial agronomic characteristic that affects crop yield and biomass. In addition to being influenced by shoot branching, it also represents a defense strategy for higher plants to adapt to their surroundings and to avoid injury [1,2]. Within plants, axillary meristem and axillary bud development depend on the coordinated regulation of numerous signal pathways, triggered by environmental stimuli (temperature, light, nutrition, decapitation) as well as endogenous factors (plant hormones, sucrose) [3,4], with plant hormones playing an essential role for this purpose. For instance, strigolactones (SLs), which are carotenoid-derived metabolites generated in shoots as well as roots, play crucial roles in plant development [5–10], and mutations in their biosynthesis or signaling genes have been shown to promote plant branching [11–16]. Similarly, gibberellins (GAs) have been known as growth regulators for almost a century. In fact, many faulty phenotypes, including germination suppression, male sterility, dwarfing and increased tillering buds, are known to occur as a result of mutations in GA biosynthesis or signaling genes [17,18]). Furthermore, GAs are generally considered to inhibit stem branching [19], as confirmed in studies on peas (*Pisum sativum* L.), rice (*Oryza sativa* L.), and *Arabidopsis* (*Arabidopsis thaliana* L.) where they negatively regulated shoot branching [20–22]. However, in perennial strawberry (*Fragaria* × *ananassa* Duch.) and some other perennial woody plants, such as *Jatropha* (*Jatropha. curcas* L.) and hybrid aspen (*Populus tremula* L. × *P. tremuloides* Michx.), this hormone could positively regulate the growth of axillary buds [23–25]. Hence, the

regulation of plant branching by GAs differs between species. Surprisingly, both GA and SL signal transduction systems use $\alpha/\beta$-hydrolase-derived receptors which are involved in the E3 ubiquitin ligase-mediated protein degradation process, thereby revealing that these pathways could have a common evolutionary basis [26]. Further studies have shown that $GA_3$ treatment could actually lower the expression of SL biosynthesis genes, thereby reducing the level of SLs in roots. Hence, GAs could dampen the biosynthesis of SLs [27–29]. The down-regulation of GA biosynthesis genes coupled with the up-regulation of GA inactivation genes in *d17* and *d14* mutants has also resulted in lower bioactive $GA_1$ content compared with wild types [30]. Altogether, the above studies suggest that both SLs and GAs play vital roles in plant branching, with potential crosstalk occurring between them during the process.

Cucumber (*Cucumis sativus* L.) is a major vegetable crop consumed fresh or processed into pickles [31]. However, it is important to physically trim unnecessary branches to improve both its quality and yield. Thus, knowledge of the regulatory mechanism for the development of lateral branches in cucumber can be essential to assist breeding programs that yield cucumbers of various branching types. Branching is a quantitative trait [32] whose molecular mechanism is not fully understood. Recently, it was found that the cucumber gene *CsBRC1* inhibited lateral branch outgrowth by directly suppressing the functions of the auxin efflux carrier CsPIN3, thus leading to auxin accumulation in the axillary buds [33]. However, only few reports are available on the interactions between SLs and GAs in cucumber. Therefore, it is important that such interactions are better studied in order to better understand branching development in cucumber.

In this study, an SL receptor gene *CsDAD2* was cloned from cucumber, with subsequent expression analysis showing that transcripts of *CsDAD2* were higher in the roots and stems of cucumber as well as in the plant's nodes following $GA_3$ treatment. In addition, overexpression of *CsDAD2* in *Arabidopsis* increased the number of primary cauline branches. The expression of *AtGA3ox2* also decreased significantly, although *AtGA2ox6* was upregulated in the nodes at the maturity stage. Furthermore, the *CsDAD2-OE* lines showed an increase in endogenous $GA_3$, with $GA_3$ treatment significantly down-regulating and upregulating *AtGA3ox2* and *AtGA2ox6*, respectively, during the seedling stage. The altered expression of GA-related genes not only suggested a higher $GA_3$ content in nodes but also that *BRC1* was maintained at low levels, resulting in increased branching in mature plants.

## 2. Materials and Methods

### 2.1. Plant Materials and Growth Conditions

The typical commercial cucumber 'Ba er te' and the wild-type *Arabidopsis* ecotype Columbia were used in this study. Plant materials were grown in pots containing black charcoal soil and under a 16 h:8 h photoperiod at 25 °C in a glasshouse. To examine the expression of SL-related genes, five-leaf cucumber plants were randomly divided into three groups: a control group sprayed with water, an A1 group sprayed with 50 mg/L of $GA_3$ solution, and an A2 group sprayed with 100 mg/L of $GA_3$ solution. Each plant had its leaves evenly sprayed on the adaxial and abaxial sides until drops of water were visible. Leaves, roots, stems, and nodes were taken 12 h after treatment for real-time quantitative reverse transcription-polymerase chain reaction (qRT-PCR), with three biological replicates set for each treatment.

### 2.2. Gene Isolation and Bioinformatics Analysis of CsDAD2

The cDNA sequence of the *CsDAD2* gene was first obtained from phytozome v12 (http://www.Phytozome.net/soybean.php (accessed on 12 December 2018) by using cDNA sequences of the *Arabidopsis* homologs as query sequences (TBLASTN) before designing primers to clone the coding sequence of the gene from the cDNA of cucumber (Table S1). The gene was then inserted into a pGM-Simple-T Fast vector (TIANGEN, Beijing, China), with the latter subsequently used to transform the *E. coli* DH5$\alpha$ competent cells. Plasmids from the recombinant cells were eventually extracted and sent to Sangon Biotech (Shanghai,

China) for identification. Pfam (http://pfam.xfam.org/search#tabview=tab0 (accessed on 15 October 2019)) was then used to determine the functional domain structure of CsDAD2. This protein's tertiary structure was obtained with the Phyre2 tool (http://www.sbg.bio. ic.ac.uk/phyre2 (accessed on 13 October 2019)) and edited with Pymol 2.4.0 software. CsDAD2 and its closest homologs from other species were selected and multiple sequence alignment was carried out using the DNAMAN software (Lynnon Biosoft, San Ramon, California, USA). Based on the results, a phylogenetic tree was finally generated in MEGA 6.0 software, using the neighbor-joining (NJ) method with 1000 bootstrap replications. The accession numbers of DAD2 selected sequences are listed in Table S2.

### 2.3. Vector Construction and Plant Transformation

Fragments resulting from the digestion of pGM-*CsDAD2* by *Kpn*I and *Pst*I restriction enzymes were ligated to pCAMBIA2300 which was previously digested with the same restriction enzymes in order to generate an over-expression construct 35S-*CsDAD2*. This recombinant vector was then transformed into *E. coli* DH5$\alpha$ competent cells for amplification before being introduced into *Agrobacterium tumefaciens* GV3101. The latter was subsequently used to infect wild-type (WT) *Arabidopsis thaliana* by the inflorescence infection method to generate *CsDAD2*-overexpression lines (*CsDAD2-OE* lines). Successfully transformed plants were screened using 1/2 MS medium plates containing 25 mg/L kanamycin and identified by PCR.

### 2.4. Treatment of Transgenic Arabidopsis

To examine the expression of GA-related genes, *Arabidopsis* from the *CsDAD2-OE* lines and wild-type ones were cultured in two forms. In the first case, the transgenic *Arabidopsis* and WT were grown in plates containing 1/2 MS medium. When the plants grew to eight leaves, whole seedlings were taken for qRT-PCR and the GA$_3$ content was analyzed by HPLC. The remaining plants were then treated as follows: (1) Control group sprayed with water. (2) C1 group sprayed with 50 mg/L of GA$_3$ solution. (3) C2 group sprayed with 100 mg/L of GA$_3$ solution. (4) C3 group sprayed with 50 mg/L of paclobutrazol (PAC) solution. (5) C4 group sprayed with 100 mg/L of PAC solution. Whole seedlings of these five groups were taken for qRT-PCR three hours after treatment. In addition, plant materials grown in pots containing black charcoal soil had their roots, leaves, first nodes, first internodes, and third nodes taken for qRT-PCR when the plants grew to maturity. The number of rosette branches and primary cauline branches was also recorded. Three biological replicates were set for each treatment.

### 2.5. Extraction and Quantification of GA$_3$ in Arabidopsis

After mixing approximately 0.1 g of the *Arabidopsis* seedlings with 1 mL of precooled reagent (methanol:water:acetic acid = 80:20:1), the mixture was allowed to leach for 12 h at 4 °C. Centrifugation was then performed at 8000 rpm for 10 min and after collecting the supernatant, the remaining residue was mixed with 0.5 mL of precooled reagent (methanol:water:acetic acid = 80:20:1). The above process was again repeated and both supernatants were combined. This was followed by the removal of any organic phase by blowing nitrogen at 40 °C. Residues were subsequently mixed with 0.5 mL of petroleum ether three times for decolorization at 60–90 °C before discarding the organic phase. After adjusting the pH to 2.8 with saturated aqueous citric acid, ethyl acetate extraction was carried out three times, with the organic phase merged together. The latter was dried by blowing nitrogen gas and after adding methanol (0.5 mL) for eddy shock dissolution, the sample was filtered.

HPLC analysis of the extracted hormones was performed using a Rigol L3000 system (Beijing Puyuan Jingdian Technology, Beijing, China). Different GA$_3$ standard solutions (0.1, 0.5, 1.0, 2.0, and 5.0 µg/mL) were prepared using methanol as the solvent, with two replicates for each standard concentration. The HPLC, equipped with a Kromasil C18 reversed-phase chromatographic column (250 mm × 4.6 mm, 5 µm), was then performed

under the following conditions: ultraviolet wavelength of 210 nm, a column temperature of 30 °C, mobile phase (A:0.1% phosphoric acid, B:methanol, A:B = 6.5:3.5), an injection volume of 10 μL, a flow rate of 1 mL/min, and a retention time of 40 min. After the baseline had stabilized, the mobile phase was run through the column and samples were added.

### 2.6. RNA Isolation and qRT-PCR

Total RNA was first extracted using an RNA pure plant kit (Kangwei Century Company, Beijing, China) according to the manufacturer's instructions before synthesizing cDNA using the First Strand cDNA Synthesis Kit (TOYOBO, Shanghai, China). The cDNA was then diluted for qRT-PCR in three biological repeats according to the UltraSYBR Mixture kit (Kangwei Century Company, Beijing, China). Selected primers for the qRT-PCR are shown in Tables S3–S4. The cycling conditions, as applied on a 7500 Real-Time PCR System (Applied Biosystems, Waltham, MA, USA), were as follows: 94 °C for 20 s, 56 °C for 20 s, and 72 °C for 30 s. In this case, *actin* was used as an internal reference for the gene expression analysis. The $2^{-\Delta\Delta Ct}$ method was eventually applied to assess the relative expression level of each gene.

### 2.7. Statistical Analysis

Statistical analysis of the data was performed using Microsoft Office Excel 2016 and IBM SPSS Statistics 20. All results were presented as the mean $\pm$ standard deviation (SD) of three replicates (n = 3). Gene expression levels were statistically evaluated by analysis of test results. Other data were statistically evaluated by analysis of variance (ANOVA), and the means were compared by Duncan's multiple comparisons. Significant differences between different treatments were determined at $p < 0.05$ and $p < 0.01$ significance levels. Charts were generated with Origin 2019, Microsoft PowerPoint, and R (version 4.2.2) software.

## 3. Results

### 3.1. CsDAD2 Cloning and Bioinformatics Analysis

The expression of the SL biosynthesis gene *CsCCD7*, receptor genes *CsD14* and *Cs-DAD2*, and branching regulatory genes *CsTCP2* and *CsTCP18* (*BRC1*) were analyzed in cucumber which had been treated with different concentrations of GA$_3$ in order to assess whether the GAs had any effects on the SLs and on the development of branches. The results showed that GA$_3$ treatment influenced the expression of both the SL-related genes and the branching regulatory genes. In particular, increased expression of *CsDAD2* was noted in leaves, stems, and nodes after two concentrations of GA$_3$ treatment (Figure S1). Therefore, *CsDAD2* was selected for subsequent experiments to explore the molecular mechanism of the interaction between SLs and GAs.

The CDS region of the cucumber *CsDAD2* gene was obtained by PCR amplification and its sequence of 908 bp encoded a protein of 276 aa, with the 29–269 aa being the functional domain of abhydrolase-6. Thus, *CsDAD2* belonged to the alpha/beta hydrolase family (Figure 1a). A total of 271 aa residues were modeled, with 100% confidence and using a single template with the highest score. The image of the tertiary structure of CsDAD2 was colored by a rainbow from the N to the C end (Figure 1b). After protein sequence alignment, it was found that the CsDAD2 protein sequence had a large number of conserved sites with homologous proteins from other species. In particular, the similarity with *Benincasa hispida* reached 87.73% (Figure 1c). A phylogenetic tree was then built using CsDAD2 and the homologous proteins from other species. In this case, it was found that this protein had a very close evolutionary relationship with *Benincasa hispida*, while being further from *Arabidopsis thaliana* (Figure 1d).

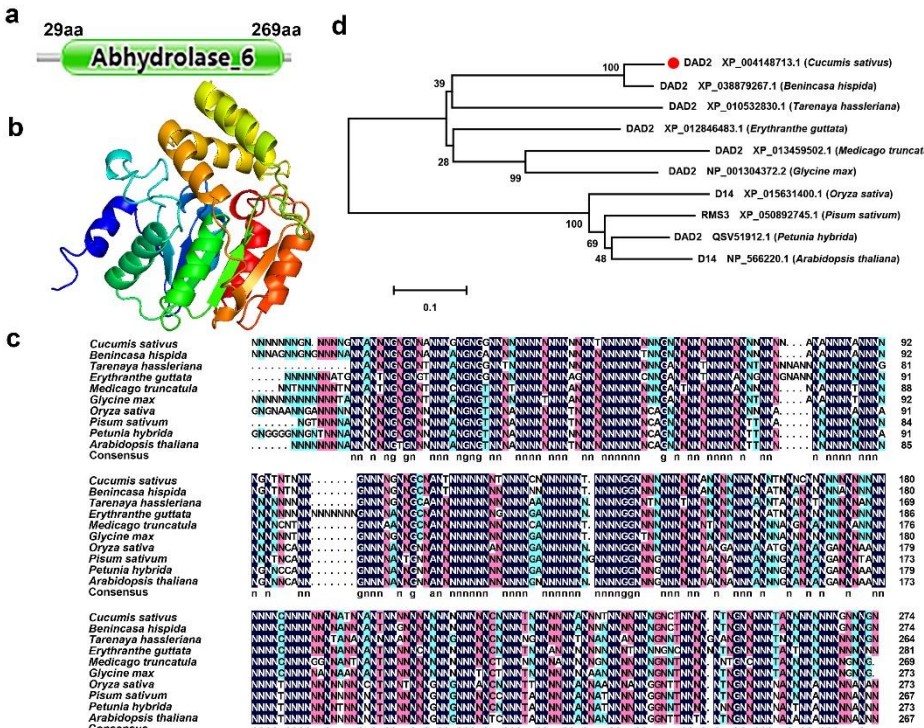

**Figure 1.** Sequence analysis of *CsDAD2*. (**a**) The functional domain of *CsDAD2*; (**b**) the tertiary structure of *CsDAD2*. Using a single template with the highest score, a total of 271 amino acid residues were modeled with 100% confidence. This picture is colored by the end of the rainbow from N to C; (**c**) sequence alignment of *CsDAD2* and its homologs. The black, pink, and blue backgrounds represent 100%, 75%, and 50% similarities; (**d**) the phylogenetic relationship between CsDAD2 and its orthologs in other plants. Supplementary Table S2 lists the accession number of DAD2.

## 3.2. Analysis of CsDAD2 Expression in Different Tissues

The expression level of *CsDAD2* in different tissues of cucumber (Figure 2a) was analyzed using gene-specific primers. The results showed that transcripts of *CsDAD2* were present in leaves, roots, stems, nodes and cotyledons, but the expression levels were higher in roots and stems, where they were 85.48- and 1.38-fold higher compared with those in leaves. On the other hand, the expression levels in nodes and cotyledons were 0.41- and 1.09-fold higher than those in leaves (Figure 2b). These results indicated that *CsDAD2* was mainly expressed in the roots of cucumber.

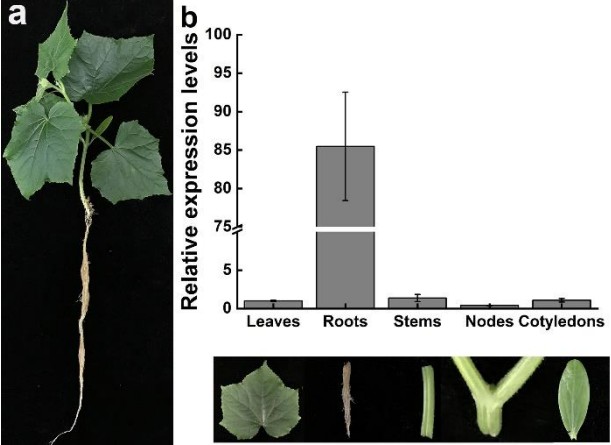

**Figure 2.** Spatio-temporal expression of *CsDAD2* in different tissues using qRT-PCR. (**a**) Cucumber plant morphology; (**b**) relative expression of *CsDAD2* in leaves, roots, stems, nodes, and cotyledons.

### 3.3. Construction of the Plant Expression Vector, Generation of Transgenic Arabidopsis, and Branching in CsDAD2-OE Lines

　　To further characterize the functions of *CsDAD2*, two T3 homozygous transgenic *Arabidopsis* strains (#1 and #2) were generated to overexpress the *CsDAD2* gene under the control of the strong constitutive CaMV35S promoter (Figure 3a,c).

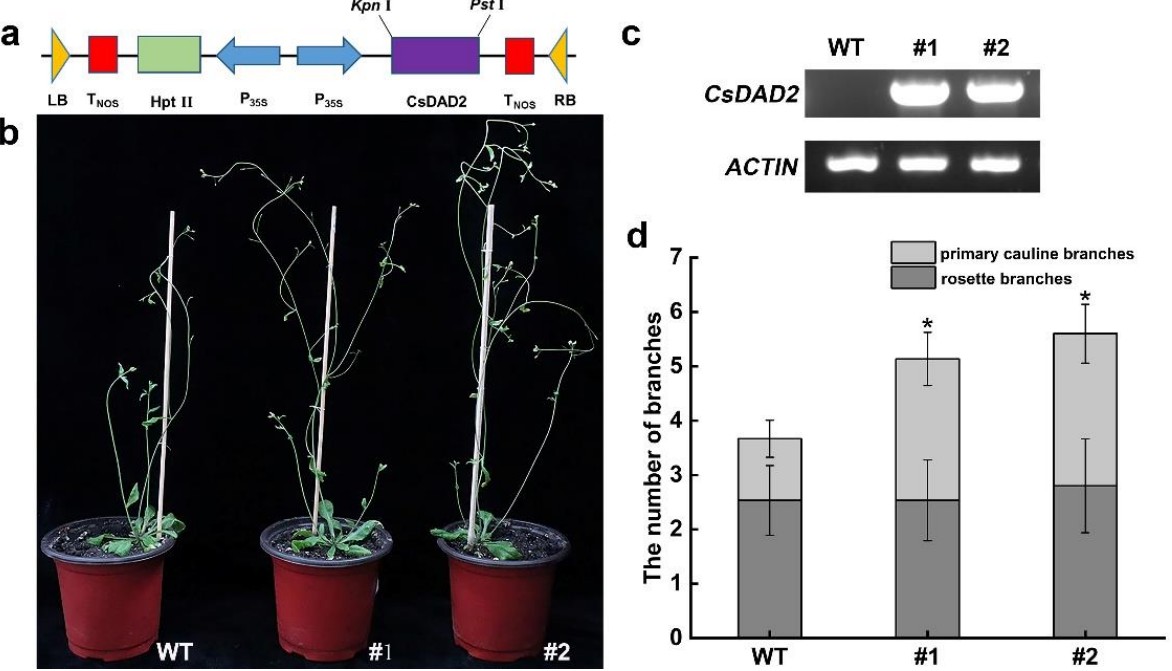

**Figure 3.** Identification and phenotypic analysis of *CsDAD2*-overexpressed *Arabidopsis*. (**a**) Schematic representation of a construct used for *Agrobacterium tumefaciens*-mediated transformation with the *CsDAD2* gene; (**b**) branching phenotypes of wild-type and representative individuals of two independent 35s::*CsDAD2* lines; (**c**) expression levels of *CsDAD2* in WT and two 35s::*CsDAD2* lines as determined by PCR; (**d**) counts of the number of rosette branches and primary cauline branches, with 15 plants used per genotype to analyze the plants' phenotypes (Student's *t* test: * $p < 0.05$).

　　The number of primary cauline branches in the two transgenic lines increased significantly compared with WT; the increase in line #1 and #2 was 2.6- and 2.8-fold higher than in WT (Figure 3b,d). Line #2 was selected for subsequent experiments.

### 3.4. The Expression of GA-Related Genes and the GA₃ Content in CsDAD2-OE Lines

　　The expression of GA-related and bud dormancy genes was determined in transgenic lines at the eight-leaf stage for examining the effects of the *CsDAD2* gene on GA signal transduction. A subset of genes consisting of the GA biosynthesis genes *AtGA20ox1*, *AtGA20ox2*, and *AtGA3ox2*; the GA signaling genes *AtGID1a*, *AtDELLA* genes (*AtRGA1*, *AtRGL*), and *AtSLY1*; the GA degradation genes *AtGA2ox2* and *AtGA2ox6*; and the bud dormancy gene *AtBRC1* were studied. The results showed that the gene expression for GA biosynthesis was reduced, with expression levels of *AtGA20ox1* and *AtGA20ox2* being 0.36- and 0.08-fold that of WT (Figure 4a). Similarly, transcripts of the GA degradation genes *AtGA2ox2* and *AtGA2ox6* were reduced (Figure 4b). Overall, the expression of GA signaling genes was downregulated, leading to obvious decreases in *AtGID1a* and *AtRGL1* expression (Figure 4c). In the case of the bud dormancy gene *AtBRC1*, the expression level was 0.47-fold that of WT (Figure 4b).

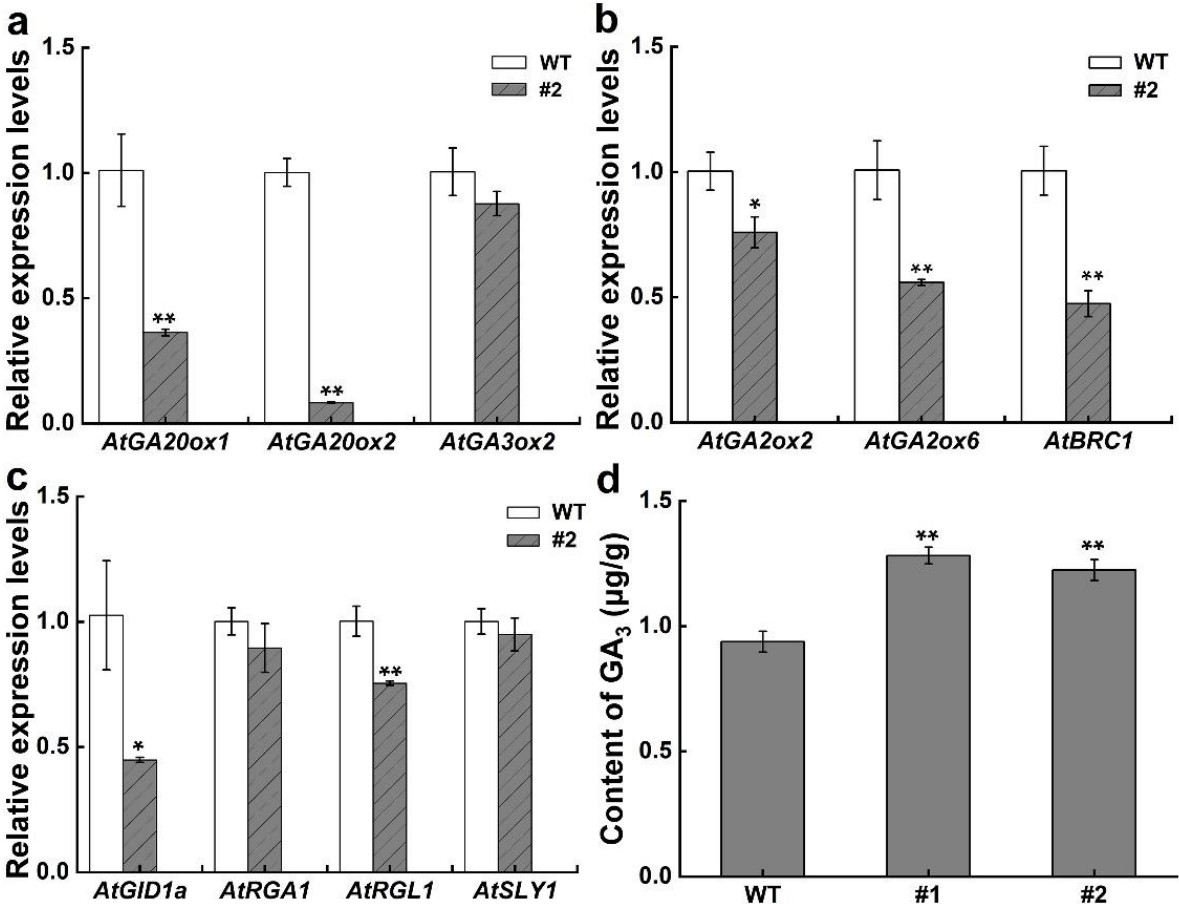

**Figure 4.** Expression of GA-related genes and the GA$_3$ content in WT as well as in *CsDAD2*-overexpressed *Arabidopsis*. Expression of GA-related genes in WT and #2 was analyzed using qRT-PCR while the GA$_3$ content in WT, #1, and #2 was determined by HPLC at the eight-leaf stage. (**a**) GA biosynthesis genes; (**b**) GA degradation genes and bud dormancy gene *AtBRC1*; (**c**) GA signaling genes; (**d**) endogenous content of GA$_3$. Data represent the mean ± standard deviation (SD) of three biological replicates (Student's *t* test: * $p < 0.05$, ** $p < 0.01$).

Furthermore, GA$_3$ content was significantly increased in the *CsDAD2-OE* line, with 0.31- and 0.37-fold increases in #1 and #2 compared with WT (Figure 4d).

### 3.5. Analysis of GA-Related Gene Expression in Different Tissues of CsDAD2-OE Lines

In different tissues of the *CsDAD2-OE* lines, the transcript levels of GA-related genes were analyzed to assess whether GA synthesis and signal transduction were regulated by *CsDAD2* locally (Figure 5). It was found that the transcripts of GA-related genes generally decreased in the roots of transgenic *Arabidopsis* compared with WT. In the case of leaves, the expression of the genes also decreased, except for *AtGA2ox6* and *AtRGA1* for which increased expression was noted. In the first nodes, the expression of the GA degradation genes *AtGA2ox2* and *AtGA2ox6* and the signaling gene *AtRGA1* increased, while the expression of all the GA biosynthesis genes as well as other signaling genes decreased. However, overall, the expression of the GA biosynthesis genes *AtGA20ox1* and *AtGA20ox2*; the GA degradation ones, *AtGA2ox2* and *AtGA2ox6*; and the signaling genes *AtGID1a* and *AtRGA1* were up-regulated in the first inter-nodes as well as the third nodes of transgenic *Arabidopsis* compared with WT (Figure 5). The transcript level of the bud dormancy gene *AtBRC1* decreased in the first and third nodes (Figure S2).

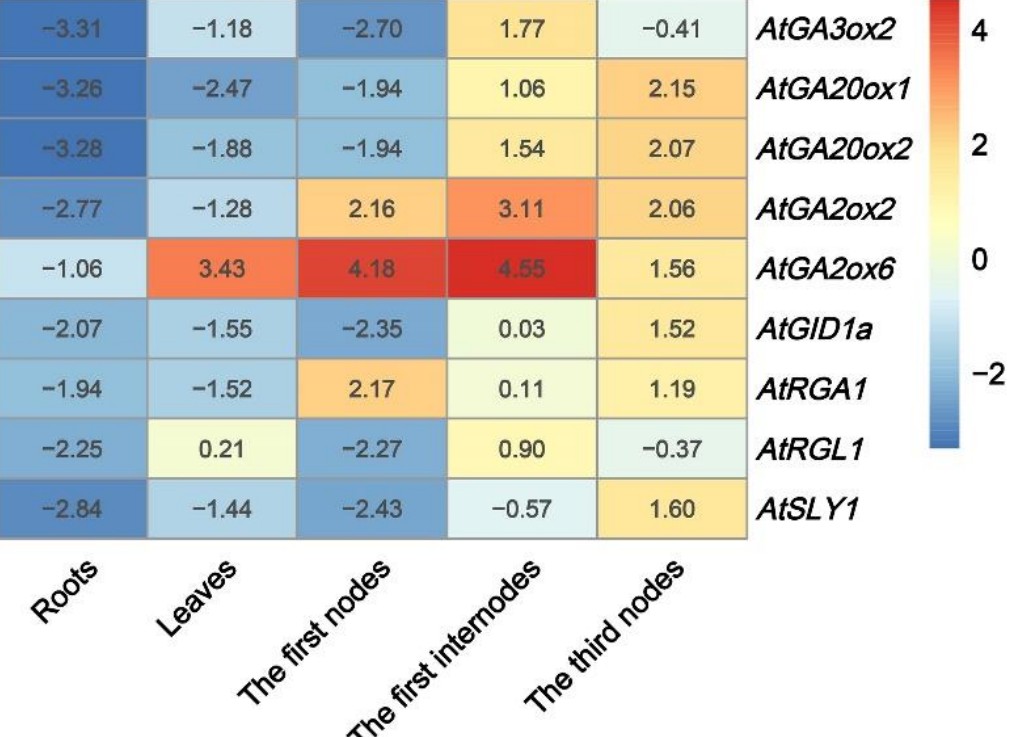

**Figure 5.** Expression of GA-related genes after *CsDAD2* overexpression in different tissues of *Arabidopsis*. The numbers represent the $\log_2$fold-change of gene expression values between #2 and WT. Positive and negative numbers respectively represent up-regulated and down-regulated gene expression in #2 relative to WT.

*3.6. Analysis of GA-Related and SL-Related Gene Expression in CsDAD2-OE Lines after GA$_3$ and PAC Treatments*

To explore the molecular mechanism of interactions between CsDAD2 and GAs further, the transcript levels of GA-related genes and SL-related genes were assayed in *CsDAD2-OE* lines after GA$_3$ and PAC (an inhibitor of GA$_3$ biosynthesis) treatment. The expression of GA synthesis genes was significantly decreased in transgenic lines treated with different concentrations of GA$_3$ compared with the control (CK/#2). Conversely, *AtGA3ox2* was significantly more expressed after PAC treatment at two concentrations (Figure 6a). Similarly, the transcript levels of the GA degradation genes *AtGA2ox2* and *AtGA2ox6* were significantly increased with the latter showing a 6.71-fold increase compared with CK at the lower concentrations of GA$_3$. However, the expression of *AtGA2ox6* decreased significantly after PAC treatment (Figure 6b). Furthermore, the expression of *AtRGA1* increased during treatment with lower concentrations of GA$_3$, while that of other signal transduction genes were suppressed. The expression of *AtGID1a* increased 2.75-fold after treatment with lower concentrations of PAC, although *AtRGA1* increased significantly after treatment with PAC; however, the gene expression for *AtRGL1* and *AtSLY1* decreased considerably (Figure 6c). These results suggested that exogenous GA and PAC may cause the feedback regulation mechanism response of the GA signal in *CsDAD2-OE* lines.

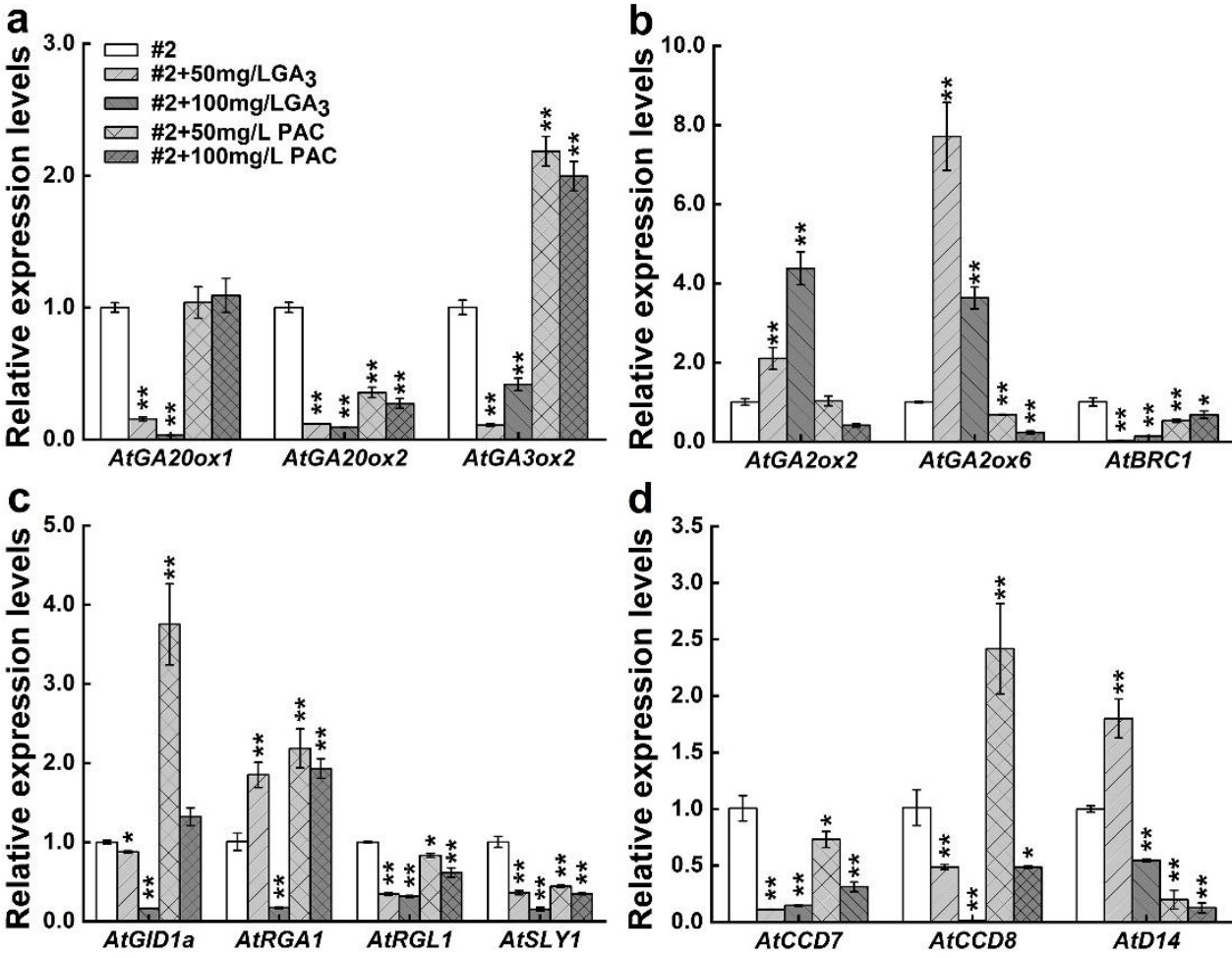

**Figure 6.** Expression of GA-related genes and SL-related ones in *CsDAD2*-overexpressed *Arabidopsis* treated with $GA_3$ and PAC. The expression of GA-related and SL-related genes was detected by qRT-PCR in *CsDAD2*-overexpressed *Arabidopsis* at the eight-leaf stage after three hours of $GA_3$ and PAC treatment (0, 50, 100 mg/L $GA_3$ and 50, 100 mg/L PAC). (**a**) GA biosynthesis genes; (**b**) GA degradation genes; and bud dormancy gene *AtBRC1*; (**c**) GA signaling genes; (**d**) SL related genes. Data represent the mean ± SD of three biological replicates (Student's *t* test: * $p < 0.05$, ** $p < 0.01$).

The expression of the SL biosynthesis genes *AtCCD7* and *AtCCD8* significantly decreased in transgenic lines treated with the two concentrations of $GA_3$ compared with CK. Moreover, the transgenic plants showed a significant increase in the expression of the SL signaling gene *AtD14* after treatment with lower concentrations of $GA_3$. However, a significant decrease was observed at the higher concentrations. The expression of *AtCCD8* increased by 1.42-fold at lower concentrations of PAC treatment, while the expression of other SL-related genes significantly decreased (Figure 6d). The expression levels of bud dormancy gene *AtBRC1* was significantly decreased after $GA_3$ and PAC treatment (Figure 6b).

## 4. Discussion

A hormone network regulates shoot branching, and in this process strigolactones, auxin, and cytokinin were found to interact, thereby leading to the proposal of two models, namely, the auxin canalization model [34–36] and the messenger model [37], both of which can be used to explain the development of shoot branches. However, GAs also have substantial regulatory effects on branching, but studies on their relationships with other hormones are relatively scarce. It was found that the genes that encode the SL receptors, *D14a* and *D14b*, were strongly upregulated by $GA_3$ and $GA_4$ in hybrid aspen [38]. Based

on the literature data, it was hypothesized that SL receptor genes could have a role in the interactions between GAs and SLs. In this study, the expression of the SL-related gene *CsCCD7* and SL receptor genes (*CsD14* and *CsDAD2*) were analyzed in cucumber after GA$_3$ treatment. Surprisingly, two concentrations of GA$_3$ significantly promoted the expression of *CsDAD2* in cucumber stems and nodes (Figure S1). Therefore, this gene was subsequently studied due to its potential role in mediating the interactions between GAs and SLs. DAD2 is a homolog of D14 and belongs to the alpha/beta hydrolase fold superfamily (Figure 1a). It can recognize SLs for hydrolysis, thereby guiding the progress of the SL signaling [39–41]. *PhDAD2* expression was also higher in the leaves and axillary buds of wild petunia compared with other tissues [42]. It was further found that *CsDAD2* expression was higher in roots compared with other tissues of cucumber (Figure 2b). Hence, the transcript levels of this gene in different tissues could be related to plant species or the developmental stage of plants. Previous reports have shown that *Atd14* mutants in *Arabidopsis* could display an increase in rosette branches compared with wild-type ones [43,44], while *hvd14* mutants in *Hordeum vulgare* produced a higher number of tillers in comparison with the wild-type parent cultivar [45]. However, in this study, it was found that the number of rosette branches remained unchanged, but the primary cauline branches actually increased in the *CsDAD2-OE* lines (Figure 3). The above results suggested that *CsDAD2* overexpression may have potential side effects, which lead to an increase in the number of primary stem branches.

Interactions through the regulation of hormone biosynthesis has been reported in some studies, although direct evidence for this has been rarely presented [46–48]. In this study, the expression of GA-related genes in *CsDAD2-OE* lines was reduced (Figure 4a–c), and the GA$_3$ content increased. This might have been because GA metabolism was at a low level, but this was facilitated by the accumulation of endogenous GA$_3$ during the seedling stage in the *CsDAD2-OE* lines. It is proposed that SLs act locally in axillary buds by upregulating the expression of the *BRC1*, which is well-established as a regulator of bud outgrowth. Indeed, *brc1* mutants display increased SL-resistance to branching [49–51]. It has been suggested that SLs promote the expression of *PsBRC1* through a signal sensed by the receptor D14 [52]. This study found that the overexpression of *CsDAD2* inhibited the expression of *AtBRC1* in *Arabidopsis* seedlings (Figure 4b), and this might be related to increased endogenous GA$_3$ content. After further analyzing the expression of GA-related genes in various tissues of the *CsDAD2-OE* lines, it was found that the expression levels of *AtGA3ox2* involved in GA biosynthesis were low in the first and third nodes, but those of the GA degradation genes *AtGA2ox2* and *AtGA2ox6* increased (Figure 5), thereby indicating that *CsDAD2* retained GA biosynthesis and promoted GA degradation in the nodes of the transgenic plants. It was shown that high expression of the *AtGA20x* gene resulted in low levels of endogenous active GAs in *Arabidopsis* and *Paspalum notatum* [53]. These results could lead to a lower GA content, but it is likely that higher content of endogenous GA could result in feedback regulation of GA production in the nodes of transgenic *Arabidopsis*. *AtBRC1* expression was lower in the first and third nodes of *CsDAD2-OE* lines compared with WT at the maturity stage (Figure S2). This result was consistent with the decreased expression of *AtBRC1* at the seedling stage, indicating that the transgenic *Arabidopsis* may also contain higher concentrations of GA$_3$ at the maturity stage. Ultimately, decreased *AtBRC1* expression led to the increase in primary cauline branches of the *CsDAD2-OE* lines (Figure 3c). Hence, the results suggested that the potential side effect of *CsDAD2* overexpression led to reduced *AtBRC1* expression.

A previous study showed that RGA would not be degraded by D14 in an SL-dependent manner. Yet, the interaction between D14 and DELLA was SL-dependent [26,54]. Based on this, it was speculated that CsDAD2 may interact with RGA. This result was confirmed in the *CsDAD2-OE* lines at the maturity stage where the overexpression of *CsDAD2* promoted the expression of *AtRGA1* in the first and third nodes (Figure 5). Therefore, it is proved that *DAD2* positively regulates the expression of *RGA1*.

Analysis of the expression of GA-related genes in *CsDAD2-OE* line seedlings indicated that $GA_3$ led to a significant decrease in the expression of the biosynthesis genes, while PAC promoted the expression of *AtGA3ox2* (Figure 6a). In addition, the GA degradation genes *AtGA2ox2* and *AtGA2ox6* significantly increased in expression after $GA_3$ treatment, and *AtGA2ox6* significantly decreased after PAC treatment (Figure 6b). These results were similar to previous reports where $GA_3$ strongly upregulated *GA2ox* genes [38,55]. The results, therefore, showed that exogenous $GA_3$ and PAC could induce feedback regulation on the endogenous content of GAs, especially through *AtGA3ox2* and *AtGA2ox6*, in transgenic plants. Changes in these two genes after $GA_3$ treatment were consistent with those in transgenic *Arabidopsis* nodes (Figure 5), suggesting that high concentrations of $GA_3$ might be present at this site. The application of $GA_3$ decreased the expression of *CsBRC1* in the *CsDAD2-OE* lines (Figure 6b). Reduced *AtBRC1* could be the cause of $GA_3$ content in the nodes of transgenic plants. However, GAs were negative regulators of rice tillering, and the expression of *TB1* (*BRC1*) was increased with the use of $GA_3$ in wild-type rice plants [18]. Therefore, the regulation of plant branching by GAs differs between species. Overexpression of *CsDAD2* is thought to impact other regulatory factors such as cytokinin, thus reducing the expression of *AtBRC1* [24,56–58]. These results suggested that a complex regulatory network could be formed to inhibit the expression of *AtBRC1* including the interaction between *DAD2* and GA, thereby improving the branching process.

In fact, $GA_3$ treatment was shown to reduce the expression of SL synthesis genes such as *OsD10/CCD8* and *OsD17/CCD7*, thereby reducing the level of SLs in rice roots [27,28]. It was also found that the expression of *AtCCD7* and *AtCCD8* decreased significantly in *CsDAD2-OE* lines after $GA_3$ treatment, thus revealing that gibberellin has an effect on SL biosynthesis in *CsDAD2*-overexpressed plants. Simply lowering the concentration of PCA further promoted the expression of *AtCCD8*, unlike the $GA_3$ treatment, whereas the expression of *AtCCD7* still decreased (Figure 6d). Therefore, *AtCCD8* and *AtCCD7* had different effects on PAC, especially in terms of the concentrations. The expression of *D14a* and *D14b* was increased in hybrid aspen fed with $GA_3$ at concentrations of 10 μm [38]. Similarly, *AtD14* expression increased significantly in *CsDAD2-OE* lines after treatment with lower concentrations of $GA_3$, although the expression significantly decreased after PAC treatment at both concentrations. However, it was significantly decreased after treatment with higher concentrations of $GA_3$ (Figure 6d). Therefore, the expression of *AtD14* in the *CsDAD2-OE* lines did display unsimilar responses at different concentrations of $GA_3$.

**5. Conclusions**

In summary, this study showed that *CsDAD2* could be involved in the interactions between GAs and SLs. These interactions were discovered using *CsDAD2*-overexpressed lines. *CsDAD2* might promote GA metabolism at a low level and then facilitates the accumulation of endogenous $GA_3$ during the seedling stage. Furthermore, *CsDAD2* overexpression might promote the accumulation of $GA_3$ content in plant nodes, and a surge of $GA_3$ leads to the decreased expression of the GA biosynthesis gene *AtGA3ox2* and the elevated expression of the GA degradation gene *AtGA2ox6*, before ultimately leading to a decrease in the expression of the branching inhibitor gene *BRC1*, thereby promoting primary cauline branches at the maturity stage.

**Supplementary Materials:** The following supporting information can be downloaded at: https://www.mdpi.com/article/10.3390/horticulturae9010023/s1.

**Author Contributions:** Conceptualization, Y.C. and Y.D.; methodology, R.P.; software, Q.L. and R.Z.; validation, Y.C. and M.L.; formal analysis, C.C.; investigation, Y.C. and X.J.; resources, X.J. and C.C.; data curation, Y.C. and X.J.; writing—original draft preparation, Y.C. and Y.D., writing—review and editing, X.J. and C.C.; visualization, Y.C. and Q.L.; supervision, X.J. and C.C.; project administration, X.J.; funding acquisition, X.J. All authors have read and agreed to the published version of the manuscript.

**Funding:** This work was supported by the Natural Science Foundation of Heilongjiang Province (LH2021-C052) and the 2021 Heilongjiang University Student Innovation Training Program (202110231012).

**Data Availability Statement:** Not applicable.

**Conflicts of Interest:** The authors declare that the research was conducted in the absence of any commercial or financial relationships that could be construed as a potential conflict of interest.

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
