# Peer review of "Cucumber Strigolactone Receptor CsDAD2 and GA3 Interact to Regulate Shoot Branching in Arabidopsis thaliana L."

_horticulturae, doi:10.3390/horticulturae9010023_

Round 1

Reviewer 1 Report

The reports on the interaction between SLs and GAs genes are limited, therefore, study theses interactions to better understand the branching development is important.

The results have theoretical and practical significance.

The experiment design is appropriate. However, it would be better if author had more information on characteristics of the CsDAD2-overexpressed transgenic plant and the results on comparison of the expression of GAs genes in over-, normal-, and weak-expressed transgenic plants with the control (non-transgenic plant).

The figures are appropriate.

The English is appropriate.

The cited references are relevant. But are not up to date (very few references in last five years). Therefore, the related and updated references should be added.

The manuscript can be accepted for publication after reversion based on reviewer's comments and suggestions.

Reviewer 2 Report

Dear Authors,

Some suggestions and corrections were made in the article.

The following aspects are brought to the attention of the authors.

1.

without space between bibliographic sources

eg

Page 1, row 38

”[11,12,13,14,15,16].” instead of ”[11,12,13,14, 15, 16].”

2.

If possible, it is recommended to revise the font size in the figures compared to the text settings in the article content.

Eg

Figures 4, 6, but also in others

3.

Italic Font Style for species name

Eg

Page 9, row 334

"Hordeum vulgare" instead of "Hordeum vulgare"

4.

References

The entire References chapter needs to be revised according to the Instructions for Authors, and Microsoft Word template, Horticulturae journal.

Author 1, A.B.; Author 2, C.D. Title of the article. Abbreviated Journal Name Year, Volume, page range

Include the digital object identifier (DOI) for all references where available.

YearBold Font Style

eg

Page 11, row 431

2010” instead of “2010”

VolumeItalic Font Style

eg

Page 11, row 431

15” instead of “15”

It is recommended to review each bibliographic source and correct it, if necessary

Reviewer 3 Report

This manuscript showed that CsDAD2 could be involved in the interactions between GAs and SLs. CsDAD2 overexpression might promote the accumulation of GA3 content in plant nodes and increased significantly the number of primary cauline branches at the maturity stage. AtBRC1 decreased at the seedling stage as well as at maturity stage of CsDAD2-OE lines.

1.    In the abstract, the content is not concise enough in line 14-16, and the description of the sentence is not clear in line 22-24.

2.    line 178, there are many important genes that have not been detected in the SL pathway, such as CCD8, MAX2, D53, etc. It was not determined whether those genes were more important key factors involved in GAs and SLs interactions

3.    It was previously reported that D14 and DELLA can interact, so can DELLA interact with DAD2 in cucumber.

4.    There are two SL receptor genes in cucumber, D14 and DAD2. What are the differences between them and which gene plays a major role in the SL signaling pathway.

5.    In Figure 4, check the level of other forms of GA, such as GA1 and GA7.

6.    line 410, there are errors in the sentence.

7.    In figure S2, how you measure CsBRC1 in Arabidopsis transgenic line?
